# Effect of Food with Low Enrichment of N-3 Fatty Acids in a Two-Month Diet on the Fatty Acid Content in the Plasma and Erythrocytes and on Cardiovascular Risk Markers in Healthy Young Men

**DOI:** 10.3390/nu12082207

**Published:** 2020-07-24

**Authors:** Martin Jaček, Dana Hrnčířová, Jolana Rambousková, Pavel Dlouhý, Petr Tůma

**Affiliations:** Department of Hygiene, Third Faculty of Medicine, Charles University, Ruská 87, 100 00 Prague 10, Czech Republic; martin.jacek@lf3.cuni.cz (M.J.); dana.hrncirova@lf3.cuni.cz (D.H.); jolana.rambouskova@lf3.cuni.cz (J.R.); pavel.dlouhy@lf3.cuni.cz (P.D.)

**Keywords:** n-3 polyunsaturated fatty acids, omega-3 diet, body composition, blood coagulation, blood lipids, erythrocytes, fatty acids spectrum

## Abstract

Polyunsaturated fatty acids of the n-3 series (n-3 PUFA) exhibit a number of favorable effects on the human organism and it is desirable to increase their intake in the diet. For this purpose, flaxseed oil was added to a chicken-feed mixture for the production of meat and eggs. The content of n-3 PUFA in the obtained meat was increased from 250 mg (reference value) to 900 mg in 100 g of meat and from 110 mg (reference value) to 190 mg in 100 g of whole egg; the enriched products are designated as omega-3 meat and omega-3 eggs. Omega-3 meat and eggs were subsequently fed for a period of eight weeks in an amount of 480 g of meat and four eggs (228 g netto) a week to a group of 14 healthy volunteers, whose body composition parameters were measured and blood was analyzed biochemically to determine blood lipids, coagulation parameters, plasma, and erythrocyte fatty acid spectrum composition. A control group of 14 volunteers was fed normal chicken and eggs in the same regime. The performed dietary intervention increases the intake of long-chain PUFA (LC-PUFA) by 37 mg per day, which represents 7–15% of the recommended daily dose. The performed tests demonstrated that the consumption of omega-3 enriched meat and eggs significantly increases the content of n-3 PUFA in the erythrocytes, which are a long-term indicator of fatty acid intake. This intervention has no demonstrable effect on the basic body parameters, such as body weight, fat content, Body Mass Index (BMI), and also on the plasma cholesterol level, high-density lipoprotein (HDL), low-density lipoprotein (LDL), blood clotting and inflammation markers, and omega-3 index.

## 1. Introduction

It is known that polyunsaturated fatty acids of the n-3 series (n-3 PUFA) demonstrably reduce the plasma levels of total cholesterol and triacylglycerols (TAG) in humans [1,2]. Both n-3 and n-6 PUFA play an important role in the formation, development, modulation, and stopping of inflammatory processes.

N-3 PUFA include α-linolenic acid (ALA), which is essential for humans; however, ALA received in the diet can be desaturated and elongated to long-chain PUFAs (LC-PUFA), such as eicosapentaenoic acid (EPA) and docosahexaenoic (DHA). The effectiveness of converting 18-carbon ALA to 20-carbon EPA in healthy young men is approximately 8% [3,4]. This conversion is higher in women and corresponds to 21% [4]. Most studies summarized in the article of Domenichiello et al. [5] demonstrate that less than 1% of ALA is converted to DHA. It thus follows that direct consumption of EPA and DHA is the most effective means for increasing the content of health-giving n-3 LC-PUFA in the lipids of the blood plasma and phospholipids of the cell membranes.

Metabolic syndrome is associated with obesity, dyslipidemia, hypertension, and impaired glucose tolerance, and is thus closely connected with type 2 diabetes (T2D). Replacing saturated fatty acids (SFA) by mono-unsaturated FA (MUFA) can demonstrably reduce the risk of development of metabolic syndrome [6]. Especially n-3 PUFA have the greatest ability to reduce the TAG level and total serum cholesterol (S-chol), increase the level of serum high-density lipoprotein cholesterol (S-HDLC) and activate the lipid metabolism [6,7]. Impaired glucose tolerance, as a risk factor of insulin resistance, is one of the connecting links and simultaneously the first stage of T2D. According to the meta-analysis of Qian et al. [8], PUFAs lead to a reduction of fasting glycemia by 0.87 mmol/l. Also, the effects of n-3 PUFAs on platelet function have been studied intensively [9,10]. Platelet aggregation assays are performed with agents that physiologically activate platelets in vivo, for example, adenosine diphosphate (ADP), arachidonic acid, collagen, and epinephrine. The results of clinical studies of the effect of n-3 PUFA on platelet aggregation are ambiguous. Meta-analysis of nine studies [10] demonstrated that daily supplementation with n-3 PUFAs significantly reduces ADP-induced platelet aggregation compared with a placebo with a trend towards a decrease in collagen-induced aggregation, but not in arachidonic acid-induced aggregation.

ALA occurs primarily in flaxseed oil with about 40% content of ALA and, to a lesser degree, in canola oil with a 10% content, as well as in soya oil and walnuts. The main sources of EPA and DHA are saltwater fish [11,12,13,14], krill [15], and microalgae [16,17,18]. Cardoso et al. give the amount of DHA present in Western food as approximately 100 mg/day [11]. At the same time, the recommended value for consumption of EPA and DHA for the European adult population is in the range 250–500 mg/day [11,12,19,20]. Consequently, in addition to food supplements containing fish oil, other means are sought to supplement the n-3 LC-PUFA content in our diets. The smell of fish oil containing approximately 18% EPA and 12% DHA, or its concentrate [21], makes it quite unacceptable for many individuals.

Another possibility represents stearidonic acid (SDA; C18:4n3), from which EPA is synthesized more effectively than from ALA. Prasad et al. mentions 14–16% conversion of SDA to EPA [22] and Bowen et al. even 33% [23]. A high amount of SDA is reported in hemp oil, blackcurrant oil, and echium oil, however its conversion to DHA is low [22]. Another option lies in the breeding of farm animals fed a mixture containing ALA derived from natural plant sources, such as flaxseed and flaxseed oil, where it is assumed that ALA is converted to higher n-3 LC-PUFA in these animals. Cortinas et al. showed a 16-fold increase in ALA in chicken breast meat to a level of 4.1 g/kg and an almost 30-fold increase in ALA in thigh meat with skin to 31.4 g/kg, when applied approximately 7% flaxseed oil and 2% fish oil into feed mixture [24]. The increase in EPA and DHA, derived from the fish oil or from the ALA conversion, reached 2.39 g/kg (340% increase) for EPA and 1.13 g/kg (66% increase) for DHA; 0.3 g/kg EPA (130% increase) and 0.4 g/kg EPA (40% increase) were measured in the chicken breast. The greater part of the FA received is thus stored in the skin and subcutaneous fat. Zuidhof et al. documented triple levels of ALA and EPA in the meat of broilers using a feed mixture containing 10% flaxseed, while DHA level was unchanged [25]. The enrichment of eggs with n-3 PUFAs using various oilseeds and fish oil is summarized in a review by Fraeye et al. [26]. The 2% addition of flaxseed oil in the feed mixture reached 15 times the original level of ALA (14.88 mg/g of yolk), EPA increased from zero level to 0.37 mg/g of egg yolk, and DHA is more than doubled to 6.49 mg/g of yolk [27]. In another study, Benavides reported an 11.8-fold increase in ALA in eggs using 10% flaxseed in the feed mixture [28]. Some algae that directly produce LC-PUFA can also be used to prepare feed mixtures [16,17,18,29,30]. In this way products such as chicken and eggs enhanced in n-3 FA can be obtained directly from these animals. Disadvantages lie in the lower organoleptic quality and stability of the obtained products against spontaneous oxidation [20], which is suppressed by the addition of antioxidants, such as vitamin E to the feed mixture. Pork [31], beef [32], and carp meat [33] with elevated contents of n-3 FA can be obtained similarly.

The increased demand for enriched products led to the idea of developing basic commonly consumed foods that would contain naturally higher contents of PUFA and LC-PUFA, i.e., chicken and eggs, hereafter designated in this article as “omega-3 meat” and “omega-3 eggs”. In the Czech Republic, eggs are considered a basic food with an annual consumption of 14.6 kg brutto (263 eggs) per person in 2018; the annual consumption of chicken in the Czech Republic is 28.4 kg per person according to the Czech Statistical Office [34]. RABBIT Trhový Štěpánov Inc. (Trhový Štěpánov, Czech Republic) produces omega meat and eggs by adding flaxseed with high ALA content directly to the feed mixtures of chickens and laying hens. The submitted study is part of an extensive project concerned with the development of feed mixtures, the production of omega-3 chickens and eggs, and their wide distribution to the retail network in the Czech Republic. Because of their great nutrient value, the successful production of omega-3 meat and eggs would represent a major potential that would find its place on the market even if the retail prices of food were to increase. Here we deal with the aspect of the effect of eight-week consumption of omega-3 eggs and meat on selected biochemical and hematological blood parameters, body composition measured by the bio-impedance method, and the complete spectrum of fatty acids (FA) in the blood plasma and in the phospholipids of the erythrocyte membrane. The blood plasma lipids provide information on the actual lipid metabolism, while the membrane phospholipids provide long-term information on lipid consumption and metabolism.

## 2. Materials and Methods

### 2.1. Production of Omega-3 Meat and Eggs

RABBIT Trhový Štěpánov Inc. provided for the production of omega-3 chicken and eggs. Enrichment of foodstuffs in n-3 PUFA is based on the addition of flaxseed oil, as a natural source of n-3 PUFA, to the feed mixtures. The addition of flaxseed oil was optimized and finally an amount of 2% wt. flaxseed oil was added to the mixture for feeding chickens to produce omega-3 meat and 1% wt. flaxseed oil was added to the mixture for feeding laying hens for production of omega-3 eggs. The control groups were fed the standard feed mixture, in which soya is the source of lipids, once again in an amount of 2% wt. for the production of meat and 1% wt. for the production of eggs. Further addition of flaxseed oil to the feed mixture causes diarrhea in chickens and laying hens and is unfeasible from a feed point of view. The meat and eggs were analyzed for their contents of n-3 and n-6 PUFA at the Department of Food Analysis and Nutrition of the University of Chemistry and Technology in Prague using gas chromatography (GC) technique (Table 1).

The addition of flaxseed oil to the feed mixture was manifested in an increase in LC-PUFA from 19.9 mg to 53.8 mg per 100 g of meat (170% increase), and from 70 mg to 110 mg per 100 g of egg (57% increase). The overall amount of n-3 PUFA in the meat was changed from 250 mg to 900 mg in 100 g of meat (260% increase) and from 110 mg to 190 mg per 100 g of whole egg (73% increase). The increase in n-6 PUFA in the meat and eggs is substantially less and the thus-produced foodstuffs can be designated as omega-3 meat and eggs.

### 2.2. Dietary Study

This was a randomized study and subjects were blinded to the n-3 FA intervention. A total of 28 healthy 18–25-year-old men, students of Charles University, Third Faculty of Medicine, were recruited into the study. The volunteers were divided into two groups of 14 individuals each, and further designated as the control group (*n* = 14) and the omega-3 group (*n* = 14). Participants did not suffer from any chronic condition or take any medication, didn’t undergo any restrictive diet and did not engage in any extremely intense physical activity (defined as more than 12 h per week). Basic anamnestic data, nutritional habits, and lifestyle behaviour were self-recorded by participants in a questionnaire. All the participants were informed in detail about the study and signed a written consent. The study was approved by the Ethical Committee of the Third Faculty of Medicine (Charles University, Prague, Czech Republic), head of review board Dr. Marek Vácha, PhD. All participants were informed about the study and planned examinations and signed informed consent.

The set of volunteers was randomly divided into two groups; one received an experimental diet containing the enriched eggs and chicken and the other received a control diet. The experimental and control diets were administered four times a week in the regime of one egg (57 g netto) and 120 g of meat per serving. The control diet was prepared and administered in the same way as the experimental diet, with the difference that ordinary unenriched products (chicken and eggs) were used. According to the plan, the nutrition intervention lasted eight weeks. The meals were planned by a registered dietitian, prepared in the canteen of the Faculty Hospital Královské Vinohrady and provided to the participants under the supervision of a responsible member of the experimental team.

The average daily intake of n-3 PUFA from enriched chicken meat and eggs during the intervention is summarized in Table 2. Due to the fact that the total dietary intake of LC-PUFA was not monitored, the obtained values are related to the recommended daily intake of LC-PUFA in the Western European diet, which is 250–500 mg/day. The performed intervention represents a total daily increase of LC-PUFA intake by 37 mg compare to the control group, which is 7-15% of the recommended daily dose. The design of the food intervention was deliberately set to correspond to the usual eating habits in the population and a further increase in the consumption of eggs and meat over four eggs and four servings of meat per week) seems unrealistic in long-term practice.

The baseline diets of participants were not determined, but all the participants were instructed not to change their lifestyle, especially their dietary habits and physical activity throughout the duration of the study and not to undertake any restrictive dietary regime.

### 2.3. Anthropometric and Body Composition Data

The basic anthropometric data (weight, height, Body Mass Index (BMI)) were determined at the beginning and end of the intervention, the body composition (by the bio-impedance method) and the blood pressure were measured. Body composition values were measured by Body composition analyzer (Tanita MC 180 MA, Amsterdam, The Netherlands). All the participants were instructed not to change their lifestyle, especially their dietary habits and physical activity throughout the duration of the study and not to undertake any restrictive dietary regime. In case of acute illness, the participants were eliminated from the study.

### 2.4. Biochemical and Hematological Parameters

Blood samples were taken at the beginning and end of the intervention to determine selected biochemical and hematological parameters: blood lipids (total cholesterol, S-HDLC, serum low-density lipoprotein cholesterol (S-LDLC), triacylglycerols), hemocoagulation (aPTT, Quick test, INR, thrombocyte aggregation) and inflammatory parameters (TNF-α and IL-6). Analysis of all these parameters was performed using certified methods in the Department of Laboratory Diagnostics of Faculty Hospital Královské Vinohrady and Third Faculty of Medicine. Simultaneously, control measurements were performed of the blood pressure and heart rate.

### 2.5. Determining the Overall FA Profile in the Blood Plasma and in the Erythrocytes

Before analysis on a gas chromatograph (GC), fatty acid methyl ester (FAME) derivatives of all the samples were prepared. The FA profile in the plasma was determined using 200 µL of blood plasma, 100 µL of inert standard solution (IS) C13:0 (43 μg) and C17:0 (41 μg), and 4.2 mL of the derivatized mixture containing methanol/toluene/acetylchloride (3.2/0.8/0.2; *v*/*v*/*v*). Esterification was performed in a closed test tube for 1 h at 100 °C. Cooling was followed by neutralization with a 12% wt/*v* solution of K_2_CO_3_; the mixture was shaken for 10 min and centrifuged. The upper organic phase was then pipetted into a vial with an insert and 1 µL of sample was used for the analysis.

The FA composition of the erythrocyte membranes was determined by a slightly modified procedure according to Rose and Oklander [35]. Approximately 8 mL of full blood taken with ethylenediaminetetraacetic acid (EDTA) was centrifuged at 1780 g for 5 min and, after removing the plasma, the obtained erythrocytes were rinsed with 3 mL of physiological solution. The test tube was then shaken for 2–3 min, centrifuged at 1780× *g* for 4 min and the physiological solution was subsequently drawn off; this procedure was repeated four times. Subsequently, 1 mL of the rinsed erythrocytes was transferred to a closable test tube, 100 μL of methanol containing IS C13:0 (46 μg) and C17:0 (47 μg) were added and the solution was again shaken with 5 mL of isopropanol on a vortex shaker. The test tube was then shaken for 1 h; then 3.5 mL of chloroform was added and the mixture was again shaken for 1 h. This was followed by short centrifugation and the final extract was filtered and evaporated to dryness under a nitrogen stream at a temperature of 35 °C. The obtained lipids were stored in a freezer at −25 °C until esterification, which was performed by the same derivatization of the mixture as for the blood plasma.

All the FA determinations were performed on a GC-17A Version 3 gas chromatograph (Shimadzu, Kyoto, Japan) fitted with an AOC-20i autosampler modified for 15 samples with a standard flame ionization detector (FID). The temperatures of the injection port and FID were 250 °C and 260 °C, respectively. The basic FA profile was determined using a Stabilwax 15 m × 0.25 mm × 0.1 µm column (Restek, Bellefonte, PA, USA) and the carrier gas was helium with a flow rate of 1.2 mL/min. The temperature program began at 120 °C, followed by a gradient of 10 °C/min to 190 °C and a final gradient of 40 °C/min to 250 °C, which was maintained for 15.4 min. The splitting ratio for analysis of total FA and phospholipids from the erythrocytes was 1:60 and 1:30. The limit of quantification (LOQ) for the individual FA of the employed GC-FID technology varies in the range 1.7–8.3 μg/mL of blood plasma or erythrocytes.

The *t*-test was used for statistical processing of the set of data and the Excel program was employed to calculate parameter *p.* A value of *p* < 0.05 was taken as a statistically significant difference.

## 3. Results and Discussions

### 3.1. Physical Parameters and Basic Biochemical, Hematological, and Immunological Blood Analysis

The performed eight-week intervention with n-3 PUFA among healthy volunteers did not affect any of the measured anthropometric and body composition parameters, such as body height and weight, amount of body fat, non-fat (fat free) and muscle mass, total body water (%), or BMI, see Table 3. No statistically significant difference was found between the control and intervention group and similarly no longitudinal effect was found for eight-week regular consumption of chicken and eggs within the two groups. The energy values of the control and intervention diets were identical and the actual elevated intake of n-3 PUFA was not manifested.

Similarly, the performed intervention with elevated intake of n-3 PUFA did not demonstrate a statistically significant effect on the levels of total serum cholesterol (S-chol) and its fractions S-HDLC and LDLC, or on the serum triacylglycerol (S-TAG) level, whose values varied within the physiological limits (Table 3). For S-chol and its fractions, similar to S-TAG, no statistically significant difference was found between the control group and the omega-3 group following eight-week intervention with an elevated intake of n-3 PUFA; similarly, no longitudinal shift was observed within the individual groups. Cholesterol is ingested with the food and is simultaneously synthesized endogenously from acetyl-coenzyme-A; one of its main sources is β-oxidation of fatty acids. However, it follows from this study that elevated intake of n-3 PUFA does not affect the serum level of cholesterol and its fractions.

In the subgroup analyses, no significant effects of n-3 PUFAs on platelet function were seen in healthy participants when the participants were treated with higher doses of n-3 PUFAs ≥ 1.83 g/day or when the treatment lasted longer than eight weeks. In a study by McEwen et al., [36] four-week n-3 PUFA supplementation reduced the thrombotic potential in healthy subjects. Our study, similar to that of Cottin in 2016 [37] and a great many other studies [38,39], did not demonstrate any effect of nutrition intervention on the blood clotting parameters [37] amongst healthy individuals. The differences found were mostly caused by a zero standard deviation (SD) value.

The anti-inflammatory effect of n-3 PUFA is connected with inhibition of the NF-κB factor and reducing the levels of the interleukins IL-1, IL-6, and TNF-α. This study monitored only two immunological factors, IL-6 and TNF-α, for which no statistically significant difference cause by elevated intake of n-3 PUFA was found (Table 4). This could be caused by the fact that the anti-inflammatory properties of n-3 PUFA were demonstrated most in studies on older, hypertriglyceridemic or diabetic patients with elevated inflammatory markers, while the cytokine level amongst healthy individuals seem to relatively insensitive to n-3 LC-PUFA [40].

### 3.2. Detailed Profile of Fatty Acids in Blood Plasma and Membrane Erythrocytes

The FA profile can be monitored in the blood plasma, in the erythrocytes, and in the fat tissue. The FA content in the blood plasma is bound to the current intake and reflects the FA intake over the last few days [41]. The FA content in the erythrocyte membranes is a suitable long-term marker providing information for a period of approximately 120 days [42,43], which reflects well the FA content in other tissues, such as the liver, heart, and kidneys [21]. Nonetheless, based on the results for rats, the level in the erythrocytes need not reflect the DHA concentration in the brain [5]. The fat tissue is then a matrix in which the intake of the given FA can be followed over the longest time [43].

Monitoring FA in the blood plasma revealed an elevated ALA level in the control group before commencing the intervention (*p* = 0.01), while this difference was not observed after the intervention (Table 5). Demonstrably, the ALA level increased in the omega-3 group from the original values of 14.2 to 22.1 µg/mL (*p* = 0.00), which could be caused by the increased intake in the food, where the tested group had a three-times higher intake of n-3 PUFA consisting of 86% ALA. The plasma level of EPA and DHA did not increase during the intervention. An increase in the plasma level of oleic acid was also observed during the intervention, where its content in the omega-3 group increased from the initial level of 517 to 625 µg/mL (*p* = 0.03), which also caused a demonstrable increase in the total MUFA level (*p* = 0.04). Marginally, mention can also be made of the elevated level of C20:1n9c in the control group at the start of the intervention, which was not observed at the end of the experiment; however; similarly as for ALA, the C20:1n9c level increased during the intervention in the tested omega-3 group.

Analysis of the FA profile in the erythrocytes yielded more interesting results (Table 6). Before commencing the intervention, no statistically significant difference was observed between the two groups for any FA. At the end of the intervention, the omega-3 group had a demonstrably higher value for ALA, and also for linoleic acid, C20:0, C20:1n9c and total SFA content. When we compare the effect of regular eight-week consumption of chicken and eggs on the FA composition in the erythrocyte phospholipids within the individual groups, a statistically significant difference is found for a number of FAs. The decrease in the SFA and MUFA values in both groups should be emphasized, along with the increase in the n-3 PUFA values in the omega-3 group caused by the increase in DHA and EPA (see Figure 1). The greater intake of ALA by the test group thus did not lead to greater incorporation into the membrane erythrocytes; however, in spite of the above-mentioned low conversion level, it was reflected in the LC-PUFA that were formed by elongation and desaturation of ALA.

The sum of the percentage contents of these FA and DHA in the erythrocytes is called the omega-3 index [44]. As a result of correlation of the contents of these FA in the erythrocytes and the heart muscles, this is frequently employed to evaluate the risk of heart disease [44,45]. Its level increased in both of the monitored groups, from 4.14% to 5.41% in the control group and from 4.01% to 5.40% in the omega-3 group. From this point of view, both types of food led to an improvement in the lipid profile of the subjects and reduced the danger of cardiac arterial disease, where the initial value was slightly higher for the control group.

An increase in the omega-3 index, as well as a decrease in SFA and MUFA over the eight-week period was observed in both groups with no statistically significant difference between them. This is likely a consequence of a control diet as the eight-week intervention improved eating habits in both groups which was reflected in a lower intake of SFA, thereby increasing omega-3 index. In order to achieve a more significant effect on the monitored parameters, a higher intake of these fortified foods would probably be necessary; the question is how it is realistic in practice, as well as their long-term regular intake in the diet. Further research is needed to verify this assumption.

## 4. Conclusions

The addition of n-3 PUFA-rich flaxseed oil to the feed mixture demonstrably increases the content of especially ALA, and also of EPA and DHA in chicken and eggs, and is an effective technology for general production of n-3 FA foodstuffs. Regular eight-week consumption of omega-3 eggs and meat causes demonstrable changes in the erythrocyte phospholipid membranes, which are a long-term indicator of FA intake in foodstuffs. Amongst short-term indicators, such as the plasma FA level, an elevated ALA level caused by the ingestion of the enriched food was found. While eight-week intake of omega-3 foods was manifested in an increase in the levels of some n-3 PUFA in the blood and erythrocyte membranes, no effect was found on the basic body parameters, such as body weight, fat content, BMI, and also on the plasma cholesterol level, HDL, LDL, blood clotting, and inflammation markers.

Replacing standard soy with flaxseed oil as a source of lipids in feed mixture for fatty chickens and laying hens will ensure an enrichment of n-3 PUFAs from 250 mg to 900 mg/100 g of meat, and from 110 mg to 190 mg/100 g of eggs. With normal consumption of four servings of fortified meat and four whole eggs per week, this intervention increase the total LC-PUFA intake by 37 mg per day, which represents 7-15% of the recommended daily dose. This intervention has no demonstrable effect on the basic body parameters, such as body weight, fat content, BMI, and also on the plasma cholesterol level, HDL, LDL, blood clotting, inflammation markers, and omega-3 index.

## Figures and Tables

**Figure 1 nutrients-12-02207-f001:**
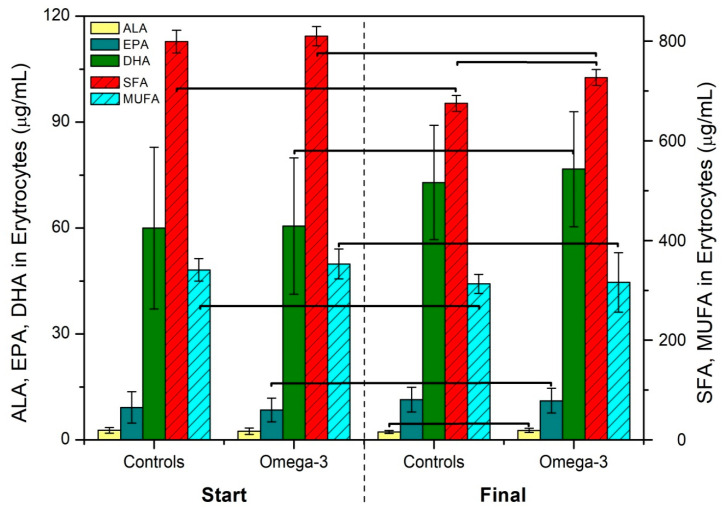
Comparison of the difference in the contents of ALA, EPA, DHA, Σ SFA and Σ MUFA during the intervention. Statistically significant differences between the individual groups with *p* < 0.05 are designated by a bracket. α-linolenic acid (ALA), eicosapentaenoic acid (EPA), docosahexaenoic (DHA), saturated fatty acids (SFA), monounsaturated fatty acids (MUFA)

**Table 1 nutrients-12-02207-t001:** PUFA composition of omega-3 chicken and eggs.

PUFA	Chicken Thigh Muscle (mg FA/100 g of Meat)		Egg (mg FA/100 g of Whole Egg)	
	Control Chicken	Omega-3 Chicken	Control Egg	Omega-3 Egg
**ALA (C18:3n3)**	210	810	40	80
**EPA (C20:5n3)**	7.3	22.3	0.0	10
**DHA (C22:6n3)**	12.6	31.5	70	100
**LC-PUFA**	19.9	53.8	70	110
**Ʃ n-3 PUFA**	250	900	110	190
**Ʃ n-6 PUFA**	2040	2280	860	1070

Polyunsaturated fatty acids (PUFA), fatty acids (FA), α-linolenic acid (ALA), eicosapentaenoic acid (EPA), and docosahexaenoic (DHA).

**Table 2 nutrients-12-02207-t002:** Average daily intake of n-3 PUFA in mg/day from chicken meat and eggs during the intervention above the baseline diet.

N-3 PUFA (mg/day)	Control Group			Omega-3 Group		
	Meat	Egg	Total	Meat	Egg	Total
**ALA**	144	13	157	555	26.1	581
**EPA**	5.0	0.0	5.0	15.3	3.3	19
**DHA**	8.6	22.8	31	21.6	32.6	54
**LC-PUFA**	13.6	22.8	36	37	36	73

**Table 3 nutrients-12-02207-t003:** Anthropometric and body composition measurements.

Parameter	Start					Final					Start/Final	
	Controls		Intervention			Controls		Intervention			Controls	Intervention
	Mean	SD	Mean	SD	*p*	Mean	SD	Mean	SD	*p*	*p*	*p*
**Height, cm**	182	6.14	181	4.96	0.79	182	6.17	181	4.96	0.82	0.98	1.0
**Weight, kg**	74.6	5.60	71.3	6.81	0.15	74.9	6.18	71.8	6.27	0.18	0.89	0.82
**Fat, %**	12.5	3.87	11.2	4.06	0.38	12.8	2.91	12.0	2.74	0.46	0.80	0.50
**Fat, kg**	9.39	3.13	8.15	3.33	0.30	9.62	2.36	8.71	2.34	0.29	0.82	0.59
**Non-fat, kg**	65.3	5.19	63.2	4.84	0.25	65.3	5.62	63.1	5.23	0.27	0.98	0.99
**Muscle, kg**	62.0	4.96	60.0	4.62	0.25	62.1	5.37	60.0	4.99	0.27	0.98	1.0
**Water, %**	63.4	2.98	65.2	4.83	0.24	63.2	2.33	63.8	2.45	0.55	0.83	0.29
**BMI, kg/m^2^**	22.6	1.42	21.7	1.57	0.10	22.7	1.29	21.9	1.25	0.08	0.85	0.74

Standard deviation (SD), body mass index (BMI)

**Table 4 nutrients-12-02207-t004:** Basic biochemical, hematological, and immunological laboratory test of blood serum.

Test	Start					Final					Start/Final	
	Controls		Intervention			Controls		Intervention			Controls	Intervention
	Mean	SD	Mean	SD	*p*	Mean	SD	Mean	SD	*p*	*p*	*p*
**Biochemical**												
**S-chol**	3.92	0.64	4.14	0.57	0.33	3.99	0.72	4.09	0.67	0.7	0.78	0.83
**S-HDLC**	1.39	0.16	1.36	0.22	0.76	1.79	1.57	1.33	0.27	0.25	0.32	0.67
**S-LDLC**	2.11	0.55	2.36	0.56	0.22	2.19	0.48	2.27	0.72	0.72	0.67	0.69
**S-TAG**	0.95	0.33	0.92	0.25	0.77	0.9	0.29	1.1	0.55	0.24	0.71	0.24
**Hematological**												
**AgRisto**	85.4	3.51	85.4	3.1	1.0	81.3	8.93	83.1	4.66	0.48	0.1	0.1
**AgADP**	80.1	7.48	79.2	3.98	0.68	77.7	4.9	78.5	6.92	0.71	0.3	0.71
**AgCollag**	82.6	2.56	81.5	3.47	0.32	78.3	5.76	80.5	4.96	0.27	**0.01**	0.51
**AgEpi**	80.2	3.74	77	13.7	0.39	78.3	4.86	78.2	11.3	0.98	0.25	0.78
**AgAra**	83.4	2.72	82.9	3.89	0.66	80.3	2.88	82	4.67	0.21	**0.0**	0.59
**QuickTest**	14.2	0.71	14.1	0.81	0.75	14.3	0.57	14.2	0.82	0.71	0.65	0.73
**Quickcontr**	12.9	0.0	12.9	0.0	1.0	13.2	0.0	13.2	0.0	1.0	**0.0**	**0.0**
**INR**	1.14	0.08	1.13	0.09	0.75	1.11	0.06	1.1	0.08	0.71	0.27	0.36
**APTTtest**	37.4	2.33	36.9	2.87	0.61	36.3	2.36	35.8	3.15	0.63	0.21	0.31
**APTTcontr**	33.1	0.17	33.1	0.19	0.49	32.5	0.0	32.5	0.0	1.0	**0.0**	**0.0**
**APTT/R**	1.13	0.07	1.12	0.09	0.7	1.12	0.07	1.1	0.1	0.64	0.66	0.65
**Immunological**												
**IL6**	2.97	3.87	0.99	0.79	0.06	1.97	3.92	2.31	3.07	0.79	0.49	0.11
**TNF-α**	1.47	0.95	1.26	1.06	0.55	1.24	0.77	1.12	0.55	0.64	0.46	0.66

Abbreviations and units: S-chol: Serum Total Cholesterol (mmol/L); S-HDLC: Serum High-Density Lipoprotein Cholesterol (mmol/L); S-LDLC: Serum Low-Density Lipoprotein Cholesterol (mmol/L); S-TAG: Serum Triacylglycerols (mmol/L); AgRisto: Ristocetine-induced platelet aggregation (%); AgADP: ADP-Induced Platelet Aggregation (%); AgCollag: Collagen-Induced Platelet Aggregation (%); AgEpi: Epinephrine-Induced Platelet Aggregation (%); AgAra: Arachidonic acid-Induced Aggregation (%); QuickTest: Quick Test (s); INR: International Normalized Ratio; APTT: Activated Parcial Thromboplastin Time (s); APTT/R: Activated Partial Thromboplastin Time Ratio; IL6: Interleukin 6 (pg/mL); TNF-α: Tumour Necrosis Factor α (pg/mL).

**Table 5 nutrients-12-02207-t005:** Plasma FA profile during intervention in µg/mL.

FA-Profile	Start					Final					Start/Final	
	Controls		Intervention			Controls		Intervention			Controls	Intervention
	Mean	SD	Mean	SD	*p*	Mean	SD	Mean	SD	*p*	*p*	*p*
C14:0	27	9.95	28.5	9.9	0.67	27.5	13.2	32.5	17.3	0.38	0.91	0.43
C16:0	566	135	569	74	0.94	559	115	637	182	0.17	0.88	0.18
C16:1n7	45.4	17.4	49.2	18	0.55	43.5	19.8	59.6	29.5	0.09	0.78	0.24
C18:0	165	36.2	169	26	0.75	167	34.8	175	29	0.53	0.87	0.56
C18:1n9c	554	141	517	75	0.36	555	142	625	180	0.24	0.98	**0.03**
C18:1n7c	44.3	12.7	44.2	8.9	0.97	46.1	12.7	48.7	9.56	0.52	0.7	0.17
C18:2n6cc	810	102	866	202	0.34	835	168	863	107	0.58	0.62	0.96
C18:3n6	11.6	5.78	10.2	4.8	0.47	10.9	4.08	10.7	4.06	0.87	0.69	0.79
C18:3n3	18.9	6.09	14.2	3.5	**0.01**	17.2	7.29	22.1	6.53	0.06	0.48	**0.00**
C20:0	7.29	1.51	8.0	1.5	0.2	7.08	1.3	7.48	1.78	0.49	0.69	0.38
C20:1n9c	7.23	0.93	6.52	0.8	**0.03**	7.21	1.28	7.49	1.36	0.55	0.96	**0.02**
C20:2n6	12.5	4.03	12	2.6	0.64	10.8	3.03	10.9	3.12	0.96	0.2	0.3
C20:3n3	45.4	12.5	43.1	13	0.63	36.4	12.1	38.5	9.02	0.6	0.06	0.26
C20:4n6	208	56.5	195	38	0.44	209	59.9	197	40.6	0.53	0.99	0.87
C20:5n3	19.6	8.83	16.3	5.8	0.22	21.7	10.5	19.3	7.58	0.47	0.56	0.21
C22:0	17.4	2.54	17.2	2.9	0.85	17.8	2.86	16.8	4.14	0.45	0.68	0.76
C24:0	18.6	2.13	18.5	2.8	0.99	21.7	20.8	17.2	2.86	0.4	0.57	0.18
C22:6n3	40.9	11.7	40.6	12	0.93	42.7	12.6	44.1	11.7	0.75	0.69	0.4
C24:1n9	34.2	7.37	32.0	4.6	0.33	31.6	10.2	31.2	9.4	0.91	0.43	0.76
Σ FA	2654	486	2656	377	0.99	2668	517	2863	496	0.29	0.94	0.19
Σ SFA	801	176	810	104	0.87	800	155	885	229	0.24	0.98	0.24
Σ MUFA	685	174	649	94.0	0.47	684	175	772	213	0.22	0.98	**0.04**
Σ PUFA	1167	162	1197	218	0.67	1184	230	1206	141	0.75	0.82	0.9
Σ n-3 PUFA	125	22.2	114	19.0	0.16	118	25.4	124	21	0.48	0.44	0.18
Σ n-6 PUFA	1043	148	1083	214	0.55	1066	215	1082	126	0.8	0.73	0.98

Fatty acids (FA), saturated fatty acids (SFA), monounsaturated fatty acids (MUFA).

**Table 6 nutrients-12-02207-t006:** FA profile in the erythrocytes during the intervention in µg/mL.

FA-profile	Start					Final					Start/Final	
	Controls		Intervention			Controls		Intervention			Controls	Intervention
	Mean	SD	Mean	SD	*p*	Mean	SD	Mean	SD	*p*	*p*	*p*
C14:0	7.54	1.06	7.6	1.27	0.89	6.88	1.26	6.95	1.31	0.89	0.13	0.16
C16:0	405	24.5	418	28.1	0.2	329	91.3	374	20.8	0.06	**0.00**	**0.00**
C16:1n7	4.71	0.65	5.65	1.81	0.07	4.71	0.73	5.41	1.21	0.06	0.97	0.67
C18:0	273	18.4	271	16	0.73	238	11.4	242	14.6	0.49	**0.00**	**0.00**
C18:1n9c	230	20.2	238	21	0.28	206	17.2	204	52	0.89	**0.00**	**0.02**
C18:1n7c	23.4	2.69	24.1	2.29	0.46	21.8	1.36	21.3	5.27	0.71	**0.05**	0.06
C18:2n6cc	184	35.7	197	26.9	0.27	173	16.8	187	17.5	**0.04**	0.3	0.23
C18:3n3	2.71	0.81	2.45	0.92	0.42	2.25	0.42	2.66	0.6	**0.04**	0.06	0.46
C20:0	8.34	1.14	8.22	1.09	0.76	7.28	0.64	7.99	1.07	**0.03**	**0.00**	0.56
C20:1n9c	4.95	0.93	4.88	0.65	0.81	3.7	1.94	5.36	1.39	**0.01**	**0.03**	0.22
C20:2n6	6.48	1.22	7.18	1.06	0.1	5.78	0.45	6.11	0.58	0.08	**0.04**	**0.00**
C20:3n3	27.4	4.99	29.7	6.28	0.27	27	4.09	28.3	4.53	0.38	0.78	0.47
C20:4n6	240	64.3	253	54.7	0.53	274	34.6	272	33.9	0.83	0.08	0.26
C20:5n3	9.18	4.45	8.48	3.35	0.63	11.4	3.52	11.1	3.48	0.79	0.13	**0.04**
C22:0	29.5	2.56	30.1	2.98	0.59	24.9	1.95	26.3	2	0.06	**0.00**	**0.00**
C24:0	75.5	12	75.4	8.56	0.98	69.1	4.4	69.5	7.41	0.83	0.06	**0.05**
C22:6n3	60	22.9	60.6	19.3	0.94	72.9	16.2	76.7	16.3	0.52	0.09	**0.02**
C24:1n9	78.4	9.35	81.1	11.5	0.49	77.3	5.14	79.7	8.9	0.36	0.68	0.71
Σ FA	1670	164	1722	144	0.35	1556	92.7	1626	107	0.06	**0.03**	**0.04**
Σ SFA	799	42.7	810	49.7	0.52	675	88.2	727	39.6	**0.04**	**0.00**	**0.00**
Σ MUFA	341	22.5	353	29.8	0.21	313	18.9	316	59.8	0.89	**0.00**	**0.03**
Σ PUFA	529	124	558	94.8	0.47	567	56	584	53.3	0.41	0.3	0.36
Σ n-3 PUFA	99.3	29.1	101	26.6	0.85	114	18.7	119	20.5	0.46	0.12	**0.05**
Σ n-6 PUFA	430	97	457	70.3	0.38	453	41.5	465	41.3	0.46	0.4	0.71

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
