# Peer review of "Effect of Food with Low Enrichment of N-3 Fatty Acids in a Two-Month Diet on the Fatty Acid Content in the Plasma and Erythrocytes and on Cardiovascular Risk Markers in Healthy Young Men"

_nutrients, 2020, doi:10.3390/nu12082207_

Round 1

Reviewer 1 Report

Thank you for your original contribution. Please see comments below. 

Please describe the dose of omega-3 provided by the study and control interventions in a more transparent manner. The reader would appreciate a more detailed description of how much EPA, DHA, and ALA the study intervention included (describe both placebo and intervention, similar reporting standards as other studies of omega-3 supplements and foods). i.e. Increasing ALA from 0.04g (40mg to 80mg) per whole egg with only 8% converted to EPA and 1% converted to DHA in young men. Authors specify that the RDA for LC-PUFA is 250-500mg per day thus at best, eating an omega-3 egg per day would increase participants daily LC-PUFA consumption by approximately 3.6mg per day (9% of 40 mg net increase of ALA). For meat, there is a relative increase of 41 mg per day of LC-PUFA consumption directly and 720 g net increase of ALA which would result in approx 65 mg increase in LC-PUFA (assuming 9% total conversion to EPA and DHA). Combined, the meat intervention could increase LC-PUFA consumption by 106 mg per day. This is orders of magnitude larger than the increase afforded by the egg intervention. This amount is comparable to the amount of LC-PUFA in some types of seafood and this should be emphasized!

Studies of supplementation with LC-PUFA often included doses of 2000-4000 mg of LC PUFA in order to see impacts on biological markers. Thus the current findings are not at all surprising. This should be highlighted to the reader. 

In the conclusion, authors should emphasize that there was no statistical effect of the study intervention on omega-3 index, the most widely used clinical marker of omega-3 status and health. Perhaps authors could speculate why the omega-3 index increased in both groups. Were baseline diets in participants measured? If not, this is a limitation as this could have confounded results. 

Author Response

Dear Reviewer,

We would like to thank you for very positive evaluation of our manuscript. All minor comments were fully considered and the manuscript has been amended accordingly. All the changes in the revised version are highlighted in red. I believe that the manuscript in the revised form will be acceptable for publication in Nutrients.

With best regards,

Petr Tůma 

Answer to reviewer:

  1. The dose of omega-3 is described in a more transparent manner. Table 2 and new paragraph is added to article; please see lines 187 – 203: “The average daily intake of n-3 PUFA from enriched chicken meat and eggs during the intervention is summarized in Table 2. In addition to direct dietary intake of EPA and DHA, 8% conversion of ALA to EPA and 1% conversion of ALA to DHA is included in the total LC-PUFA balance. Due to the fact that the total dietary intake of LC-PUFA was not monitored, the obtained values are related to the recommended daily intake of LC-PUFA in the Western European diet, which is 250 - 500 mg/day. The performed intervention represents a total daily increase of LC-PUFA intake by 74.5 mg compare to the control group, which is 15-30 % of the recommended daily dose. The design of the food intervention was deliberately set to correspond to the usual eating habits in the population and a further increase in the consumption of eggs and meat over 4 eggs and 4 servings of meat per week) seems unrealistic in long-term practice. The baseline diets of participants were not determined, but all the participants were instructed not to change their life style, especially their dietary habits and physical activity throughout the duration of the study and not to undertake any restrictive dietary regime.”
  2. A new discussion about omega-3 index is added to the text; please see lines 354 – 360: “An increase in the omega-3 index, as well as a decrease in SFA and MUFA over the 8-week period was observed in both groups with no statistically significant difference between them. This is likely a consequence of a control diet as the 8-weeks intervention improved eating habits in both groups which was reflected in a lower intake of SFA, thereby increasing omega-3 index. In order to achieve a more significant effect on the monitored parameters, a higher intake of these fortified foods would probably be necessary - the question is how it is realistic in practice, as well as their long-term regular intake in the diet. Further research is needed to verify this assumption.”
  3. The overall impact of the proposed low-level n-3 PUFA intervention on on the plasma cholesterol level, HDL, LDL, blood clotting and inflammation markers, and omega-3 index is also evaluated in the conclusion; lines 378 – 384: “Replacing standard soy with flaxseed oil as a source of lipids in feed mixture for fatty chickens and laying hens will ensure an enrichment of n-3 PUFAs from 250 mg to 900 mg/100 g of meat, and from 110 mg to 190 mg/100 g of eggs. With normal consumption of 4 servings of fortified meat and 4 whole eggs per week, this intervention increase the total LC-PUFA intake by 74.5 mg per day, which represents 15-30 % of the recommended daily dose. This intervention has no demonstrable effect on the basic body parameters, such as body weight, fat content, BMI and also on the plasma cholesterol level, HDL, LDL, blood clotting and inflammation markers, and omega-3 index.”

Reviewer 2 Report

The only way this paper can be considered in this issue is greatly reorganized as a “negative lesson” paper, showing how minor intervention with very small enrichments of ALA produce basically no results. And comment on how the diets made up and fed by the dietetic staff were the source of any positive cardiovascular health parameters regardless of whether they were ALA enriched or control.  because it has no statistical significance with respect to (especially) LCPUFA.

This paper is a for a special issue so the Introductory material in (approximately) lines 35 to 65 is not needed and will be highly redundant with other papers in the issue. The paper is based on a dietary study of n-3 enriched foods and so nearly all the Introduction should be focused on reviewing previous studies of similarly enriched foods.  n other words, for readers of special edition, basic knowledge of LCPUFA is a given; please stick to the specific topic and review it fully; do not waste words on background information. A one paragraph review of connections between n-3 PUFA intake and cardiovascular risk factors is acceptable, however it should be limited to the risk factors measured in the current study.

Line 70: Regardless of what the source may say, a total approximate intake of 100 mg/d cannot have two (eggs 5.1) or three (meat 19.6) significant figures in the intake data for individual food types! One is all that is permissible.

Flaxseed oil should be utilized rather than linseed oil for international audiences. Linseed oil refers the nonedible oil in many countries.

Do not mix omega-3, n-3, n3; pick a term and stick with it, preferably n-3.

Line 75: Either discuss and reference work regarding 18:4n-3 or omit this line.  The thought is incomplete.

Lines 187 to 194 and 214 to 222 belong in the Introduction. The statement that replacing SFA with MUFA reduces risk for metabolic syndrome required references.

Clearly there was no statistically significant enrichment in EPA and DHA in the eggs, so finding no effects on blood clotting, IL-6 and TNF-alpha, or EPA/DHA levels in plasma is not surprising.  I would not even consider this study to be a study of the effects of ingesting n-3 enriched eggs since one egg per day with so little n-3 enrichment in young men (shown to oxidize nearly all ingested ALA for energy) is irrelevant.

Lines 279 to 280: Percentages to be given to two significant figures only. Also the omega-3 index had the same increase regardless of the dietary intervention, which clearly indicates the dietary intervention itself had no effect. However the controlled diet overall did.

The minor increase in ALA probably cannot justify the cost of producing the specialized eggs and meat.  One serving of a flax or walnut oil salad dressing could have accomplished as much. This should be discussed.

Author Response

Dear Reviewer,

We would like to thank you for your valuable comments on our manuscript. All your comments were considered and the manuscript has been amended accordingly. All the changes in the revised version are highlighted in red. I believe that the manuscript in the revised form will be acceptable for publication in Nutrients.

With best regards,

Petr Tůma

Answer to reviewer:

The title is changed to: Effect of Food with Low Enrichment of N-3 Fatty Acids in a Two-month Diet on the Fatty Acid Content in the Plasma and Erythrocytes and on Cardiovascular Risk Markers in Healthy Young Men.

  1. The introductory part is completely reorganized: i) most of the original text is deleted, ii) a new paragraph on foods fortified with LC-PUFA is added with several new citation (lines 105 – 118): “Cortinas et al showed a 16-fold increase in ALA in chicken breasts meat to a level of 4.1 g/kg and an almost 30-fold increase in ALA in thigh meat with skin to 31.4 g/kg, when applied approx. 7 % flaxseed oil and 2 % fish oil into feed mixture [24]. The increase in EPA and DHA, derived from the fish oil or from the ALA conversion, reached 2.39 g/kg (340% increase) for EPA and 1.13 g/kg (66% increase) for DHA; 0.3 g/kg EPA (130% increase) and 0.4 g/kg EPA (40% increase) were measured in the chicken breast. The greater part of the FA received is thus stored in the skin and subcutaneous fat. Zuidhof et al. documented triple levels of ALA and EPA in the meat of broilers using a feed mixture containing 10 % flaxseed, while DHA level was unchanged [25]. The enrichment of eggs with n-3 PUFAs using various oilseeds and fish oil is summarized in a review by Fraeye et al. [26]. In the case of 2 % addition of flaxseed oil in the feed mixture reached Meluzi et al. 15 times the original level of ALA (14.88 mg/g of yolk), EPA increased from zero level to 0.37 mg/g of egg yolk, and DHA is more than doubled to 6.49 mg/g of yolk [27]. In another study, Benavides reported an 11.8-fold increase in ALA in eggs using 10 % flaxseed in the feed mixture [28].“; iii) the original lines 187 – 194 and 214 – 222 are moved to the introduction.
  2. The statement in the original line 70 is deleted.
  3. Flaxseed oil is utilized instead of linseed oil.
  4. New discussion and reference work regarding 18:4n-3 is added; lines 99 – 103: “Another possibility represents stearidonic acid (SDA, C18:4n3), from which EPA is synthesized more effectively than from ALA. Prasad et al. mentions 14-16 % conversion of SDA to EPA [22] and Bowen et al. even 33 % [23]. High amount of SDA is reported in hemp oil, blackcurrant oil and echium oil, however its conversion to DHA is low [22].”
  5. The labeling of fatty acids such as n-3 is united throughout the text. Only the historical “omega-3 index” or commonly used trade name “omega-3 meat”, “omega-3 egg” remains unchanged.
  6. A new discussion about omega-3 index is added to the text; please see lines 354 – 360: “An increase in the omega-3 index, as well as a decrease in SFA and MUFA over the 8-week period was observed in both groups with no statistically significant difference between them. This is likely a consequence of a control diet as the 8-weeks intervention improved eating habits in both groups which was reflected in a lower intake of SFA, thereby increasing omega-3 index. In order to achieve a more significant effect on the monitored parameters, a higher intake of these fortified foods would probably be necessary - the question is how it is realistic in practice, as well as their long-term regular intake in the diet. Further research is needed to verify this assumption.”
  7. The overall impact of the proposed low-level n-3 PUFA intervention on on the plasma cholesterol level, HDL, LDL, blood clotting and inflammation markers, and omega-3 index is also evaluated in the conclusion; lines 378 – 384: “Replacing standard soy with flaxseed oil as a source of lipids in feed mixture for fatty chickens and laying hens will ensure an enrichment of n-3 PUFAs from 250 mg to 900 mg/100 g of meat, and from 110 mg to 190 mg/100 g of eggs. With normal consumption of 4 servings of fortified meat and 4 whole eggs per week, this intervention increase the total LC-PUFA intake by 74.5 mg per day, which represents 15-30 % of the recommended daily dose. This intervention has no demonstrable effect on the basic body parameters, such as body weight, fat content, BMI and also on the plasma cholesterol level, HDL, LDL, blood clotting and inflammation markers, and omega-3 index.”

Reviewer 3 Report

The authors of this study aimed to assess how dietary intake of eggs and meat rich of omega-3 fatty acids influence body composition parameters and biochemically markers such as blood lipids, coagulation parameters, fatty acids profile.
It is a really interesting manuscript. However, I have some comments about the article:
1. In chapter Material and methods - please state on which equipment and/or method is was used to for the determination of body composition and biochemical and hematological parameters
2. Please discuss what could be the clinical utility of this result? Changes in the fatty acid profile, from the patient's point, of view, are not as significant as, e.g. improvement of coagulological parameters
3. Please explain why the concentration of SFA in RBC in the control group has decreased so significantly

Author Response

Dear Reviewer,

We would like to thank you for very positive evaluation of our manuscript. All minor comments were fully considered and the manuscript has been amended accordingly. All the changes in the revised version are highlighted in red. I believe that the manuscript in the revised form will be acceptable for publication in Nutrients.

With best regards,

Petr Tůma 

Answer to reviewer:

  1. The technical equipment and methods used for measurement of body composition, biochemical and haematological parameters are specified in more details; please see lines 207 – 208: “Body composition values were measured by Body composition analyser (Tanita MC 180 MA, Amsterdam, Netherlands).”, and 216 – 218: “Analysis of all these parameters was performed using certified methods in the Department of Laboratory Diagnostics of Faculty Hospital Královské Vinohrady and Third Faculty of Medicine.”
  2. The overall impact of the proposed low-level n-3 PUFA intervention on on the plasma cholesterol level, HDL, LDL, blood clotting and inflammation markers, and omega-3 index is also evaluated in the conclusion; lines 378 – 384: “Replacing standard soy with flaxseed oil as a source of lipids in feed mixture for fatty chickens and laying hens will ensure an enrichment of n-3 PUFAs from 250 mg to 900 mg/100 g of meat, and from 110 mg to 190 mg/100 g of eggs. With normal consumption of 4 servings of fortified meat and 4 whole eggs per week, this intervention increase the total LC-PUFA intake by 74.5 mg per day, which represents 15-30 % of the recommended daily dose. This intervention has no demonstrable effect on the basic body parameters, such as body weight, fat content, BMI and also on the plasma cholesterol level, HDL, LDL, blood clotting and inflammation markers, and omega-3 index.”
  3. The concentration of SFA in RBC and omega-3 index is discussed in new paragraph; lines 354 – 360: “An increase in the omega-3 index, as well as a decrease in SFA and MUFA over the 8-week period was observed in both groups with no statistically significant difference between them. This is likely a consequence of a control diet as the 8-weeks intervention improved eating habits in both groups which was reflected in a lower intake of SFA, thereby increasing omega-3 index. In order to achieve a more significant effect on the monitored parameters, a higher intake of these fortified foods would probably be necessary - the question is how it is realistic in practice, as well as their long-term regular intake in the diet. Further research is needed to verify this assumption.”